# Development and Psychometric Testing of a Taiwanese Team Interactions and Team Creativity Instrument (TITC-T) for Nursing Students

**DOI:** 10.3390/ijerph19137958

**Published:** 2022-06-29

**Authors:** Hsing-Yuan Liu, Su-Ching Sung, Chun-Yen Chao, Nai-Hung Chen, Hsiu-Fang Chen, Sheau-Ming Wu

**Affiliations:** 1Department of Nursing, Chang Gung University of Science and Technology, Taoyuan City 33303, Taiwan; hyliu@mail.cgust.edu.tw (H.-Y.L.); nhchen@mail.cgust.edu.tw (N.-H.C.); 2Department of Nursing, Linkous Chang Gung Memorial Hospital, Taoyuan City 33305, Taiwan; 3Department of Gerontology and Health Care Management, Chang Gung University of Science and Technology, Taoyuan City 33303, Taiwan; scsung@mail.cgust.edu.tw; 4Department of Cosmetic Science, Chang Gung University of Science and Technology, Taoyuan City 33303, Taiwan; cychao@mail.cgust.edu.tw

**Keywords:** team interaction, team creativity, scale development, nursing education, Taiwan

## Abstract

Background: How well team members work together can be affected by team interactions and creativity. There is no single instrument for measuring both variables in healthcare education settings in Taiwan. The purpose of this study is to develop an instrument to measure team interactions and team creativity for Taiwanese nursing students. Methods: A 34-item team interactions and team creativity self-report instrument was developed for nursing students in Taiwan (TITC-T). Items consisted of statements about how a participant perceived their team members’ constructive controversy, helping behaviors, communication, and creativity. Nursing students (*n* = 275) were recruited from two campuses of a science and technology university to examine the psychometric properties of the TITC-T. The reliability and psychometric properties were evaluated. Results: The Cronbach’s alpha was 0.98. The confirmatory factor analysis resulted in a one-dimensional factor structure that fit well with the model (Comparative Fit Index = 0.995, Tucker Lewis Index = 0.908, Root Mean Square Error of Approximation = 0.098). Conclusions: The TITC-T is a valid and reliable tool for evaluating team interactions and team creativity for students enrolled in nursing programs in Taiwan.

## 1. Introduction

Creativity and innovation in nursing enhances healthcare quality, reduces healthcare costs, and generates an environment structured to reduce work-related stressors and emotional exhaustion, which can reduce burnout for healthcare professionals [1]. Fostering creativity and innovation in nursing students can improve patient healthcare and outcomes [1]. Interdisciplinary collaboration among healthcare teams is also increasing, resulting in more comprehensive healthcare for the treatment of patients [2].

A systematic review on interprofessional education [3] argued that higher education must implement interdisciplinary learning opportunities. Students among different academic areas should be capable of communicating with a common language, thus affording them the opportunity to tackle more complex issues. The Word Health Organization (WHO) describes interdisciplinary education (IDE) as providing instruction from faculty with two or more professional backgrounds, which enables effective collaboration among students and improves healthcare outcomes [4]. Students involved in IDE programs not only learn relevant skills from multiple disciplines, but also learn how to communicate and collaborate. However, an understanding of interdisciplinary team interactions is critical for IDE to effectively improve student collaborations [5].

Studies have been implemented to improve training methods that can reduce barriers to and increase collaborations among team members [6,7]. However, evaluating whether team interactions and team creativity impact collaboration outcomes in nursing education have not been examined. Although many instruments have been developed to assess these variables on an individual-level, few are available for use in Taiwan that are specific for nursing education. Therefore, there is a need for the development of a valid self-report instrument for assessing team interactions and team creativity that is directed at students in nursing programs in Taiwan, especially students enrolled in IDE programs.

Boosting positive interactions and increasing creativity among team members is not only critical for the success of groups and organizations, but also a key solution to many modern challenges facing world economies, including healthcare teams [8,9]. Innovation among team members requires divergent thinking, which is more likely to occur in an atmosphere where each member feels free to express their ideas without boundaries, risks of judgment, or fear of ridicule from others [8,10]. Therefore, communication is a dominant process and an important component of team interactions that is critical to the success of healthcare teams [9,11].

Cragan et al. [12] defined four characteristics important for successful team interactions because they give the team purpose: problem solving, role playing, team building, and trust building. Constructive team interactions generate solutions to problems that have a higher level of acceptance among team members than solutions shaped by interactions that are passive or aggressive [11,13]. Team interactions, including collaboration and communication, can act as mediators between transformational leadership and performance [11,14,15].

Ilgen et al. [16] proposed that how team members interact (input) effects the product created (output). They suggested that a model of input-mediator-output-input (IMOI) could be used to understand the range of variables that are important mediators for team interactions. The IMOI model allows for a broadening of variables that can influence team interactions beyond communication and performance, described by Marlow et al. [17].

The IMOI model is the basis of several instruments developed to examine additional variables deemed important components of team interactions. Constructive controversy [18,19,20], involves open-minded discussions of incompatible views for the mutual benefit of all team members [21], and helping behaviors [20,22], consist of deliberate, straightforward efforts to assist another member of the team [23]. Assisting others has been shown to increase team interactions in general [24] and improve team creativity, which leads to the development of products that are more innovative [2,10]. Spontaneous communication [25,26] describes informal and unplanned exchanges with other members of the team outside the normal routine, which provides team members with a different perspective for problem solving [25,27]. Spontaneity helps individuals to take the initiative and be self-starting [28].

The most widely used instruments for team creativity were developed by Farh, Lee, and Farh [29], Oldham and Cummings [30] to assess team creativity for employees, and Leroy et al. [31], who assessed team creativity in the context of organizational behavior. These instruments measure collaborative behaviors, which is an important component of successful creative interdisciplinary teams and has been demonstrated to foster productivity among team members [27,30]. Instruments have also been developed to measure interaction behaviors, which can influence creativity and include constructive controversy, helping behaviors, and spontaneous communication [15,18,26]. These instruments have been adapted for studies on leadership behaviors, goal orientation, and cognitive dissonance [14,21,32].

Several qualities are important for team creativity. Effective team collaboration in an inter-professional setting has been shown to enhance patient safety and outcomes [33,34]. This requires cooperation, collaboration, and communication among members of interdisciplinary teams, and is an important component of healthcare [35]. Interactive behaviors can improve teamwork competency, which includes qualities labeled as constructive controversy, helping behavior, and spontaneous communication [36].

Creativity is considered as a collaborative behavior of successful interdisciplinary teams and has been demonstrated to foster productivity among team members [27,31]. Effective team collaboration in an inter-professional setting has been shown to enhance patient safety and outcomes [33,34]. This requires cooperation, collaboration, and communication among interdisciplinary team members, and is an important component of healthcare [35]. Interactive behaviors can improve teamwork competency, which includes qualities labeled as constructive controversy, helping behavior, and spontaneous communication [36]. Derdowski et al. [37] suggested the interactive behavior of constructive controversy can improve group performance and increase team creativity. Helping behaviors are positively related to creative innovation for healthcare teams [38]. McAlpine [39] demonstrated that spontaneous communication among team members can improve creativity by facilitating a better flow of information.

Many instruments have been developed or adapted as measures of team creativity [14,15,18,29,30,31,32]. However, these were not developed for use with nursing students in Taiwan or were administered to non-native English speakers without confirmation of the translation [14,21]. To our knowledge, there is no valid measurement instrument available for use in Taiwan that incorporates team interactions, team creativity, and competency in the healthcare setting of nursing. Therefore, because we considered it important that a self-report instrument reflects characteristics of nursing student teams, this study aimed to merge components of developed instruments for use with teams of nursing students, translate the new self-report scale into Taiwanese, and determine its validity and psychometric properties.

## 2. Materials and Methods

### 2.1. Research Design

This exploratory and cross-sectional study was performed in several steps to develop, translate, and validate an instrument to measure team interactions and team creativity for undergraduate nursing students in Taiwan.

### 2.2. Instrument Development

Although instruments for assessing team creativity and team interactions are available, they have not been developed for nursing students. Therefore, we merged relevant components from Chinese and English language scales to assess team creativity and interactions for nursing students in Taiwan. We translated all English language statements to Mandarin Chinese, the language spoken in Taiwan, using the forward translation/back translation procedure described by Brislin [40]. This translation process was used to maintain content equivalence of terminology describing team interactions in Taiwanese. English language statements were independently translated by two bilingual members of the research team who were fluent in both English and Taiwanese. An English language expert back translated the Taiwanese versions into English without any knowledge of the original English version and compared each statement to the original, to confirm semantic equivalence. The two bilingual translators confirmed the semantic equivalence of the original English statements and the back translated Mandarin Chinese statements, which indicated the final translation was satisfactory.

Team interactions assessed three components: constructive controversy [18,19,20], helping behaviors [15,20,22], and spontaneous communication [11,25,26]. The domains are measured with statements about how an individual feels about interactions among members of their team. Team creativity has been measured for information technology teams in China using a team creativity scale developed by Farh et al. for teams of employees [29], which was a modification developed by Oldham and Cummings to measure employee creativity [30]. The self-report instrument measures creative ideas, innovation, and performance and has good reliability, with a Cronbach’s alpha of 0.85 [29]. Yang et al. [15] made further modifications to measure team creativity in a research and development setting in China, again with employees rather than students; this scale has a Cronbach’s alpha of 0.95. We revised statements from the Mandarin Chinese language instruments of Farh et al. [29], Yang et al. [15], and Zhang et al. [32]. Some of the wording for the statements about team creativity was not relevant to nursing students and the challenges they would encounter as nurses. Therefore, we added statements about aspects of creativity from English language instruments that were more appropriate for an educational setting: creative performance [41] and innovation [42]. The English language statements from Shally et al. [41] and Scott and Bruce [42] were translated from English to Mandarin Chinese as described above.

We combined the items for interactions and creativity into a 34-item instrument, the Taiwanese Team Interactions and Team Creativity scale (TITC-T). A 5-point Likert scale from 1 (strongly disagree) to 5 (strongly agree) was used for the three subscales for team interactions (constructive controversy (4 items), helping behaviors (10 items) and spontaneous communication (10 items)) and 10 items for team creativity. Each item completes the statement “Members of my team…” The 34 items are shown in Table 1. The score for the three domains of team interactions is the average of the sum of all item scores in each domain and the total score for team interactions is the average of the sum of the three domains; higher scores indicate a greater perception of team interactions. The total score for team creativity is the average of the sum for all 10 items; higher scores indicate greater team creativity. The final TITC-T scale was a 34-item self-report measurement instrument for team interactions (three dimensions) and team creativity for nursing students.

### 2.3. Participants

To test the scale, nursing students (*n* = 275) were recruited from capstone courses that were part of nursing programs on two campuses of a university of science and technology in northern Taiwan. Nursing students who enrolled in the course were included if they provided signed informed consent at the conclusion of the capstone course, indicating they were willing to participate in the study.

#### The Capstone Course

The capstone course is conducted over 18 weeks by instructors who are faculty members from the programs in nursing and design. The course emphasizes working collaboratively in teams to evaluate what healthcare products could improve patient care, and then designing a prototype of the product that has the potential for being patented. Teams are comprised not only of nursing students but also students from the school of design, although the design students are not participants in the study. Students receive a midterm grade based on a written exam. The final grade consists of a group presentation at the conclusion of the course, which is presented to faculty members by each team of students.

### 2.4. Data Collection and Analysis

Data from nursing students were collected at the end of the 18-week course between October 2017 and January 2018. Two structured questionnaires were distributed to students in coded packets by a faculty member who was not part of the research team. One questionnaire collected demographic data, including age and gender; the second questionnaire was the newly developed self-report TITC-T instrument, which collected data on nursing students’ perceptions of team interactions and creativity for use in confirmatory factor analysis (CFA). Prior to analyzing the data, the faculty member checked each packet for a signed consent form. All packets contained signed consent forms, and thus, all 275 students were included in the analysis.

### 2.5. Statistical Analysis

We used SPSS version 20.0 (IBM, Chicago, IL, USA) for Windows to analyze demographics descriptive and inferential statistical analyses. Demographic and participant characteristics were analyzed via mean, standard deviation, percentage, and frequency. To explore the fit and confirm the structure of the TITC-T, we conducted CFA with structural equation modeling using AMOS 25 software (IBM, Chicago, IL, USA). All statistical tests were two-tailed, and the significance level was set to a standard of α < 0.05.

### 2.6. Ethical Considerations

Before beginning the study, approval was obtained from the university’s Institutional Review Board (IRB) (IRB201800212; IRB0C502). To maintain students’ confidentiality, data were anonymized by coding the packets and questionnaires distributed to the nursing students. 

## 3. Results

### 3.1. Participant Demographics and Mean Scale Scores

The mean age of participants was 21.4 years (SD = 0.93; range = 19 to 27 years) and 82.5% were female. The mean total scores for team interactions were 4.23 (SD = 0.48) and the three constructs were similar, as was the mean score for creativity (4.11, SD = 0.69) indicating moderately high levels of team interactions. Details of scores for each measure are shown in Table 2.

### 3.2. Confirmatory Factor Analysis

The four-factor model of the TITC-T was tested by CFA. Because the number of participants in our study exceeded 200, which can result in false positives when an χ^2^ index is used, we included the chi-squared value/degree of freedom (χ^2^/df) index [43]. In the analyzed sample (*n* = 275), indices of goodness of fit for CFA were as follows: χ^2^/df = 4.213, the goodness of fit index (GFI) = 0.832, the adjusted goodness of fit index (AGFI) = 0.973, the normed fit index (NFI) = 0.985, the incremental fit index (IFI) = 0.995, the comparative fit index (CFI) = 0.995, the Tucker–Lewis index (TLI) = 0.908, and the root mean square error of approximation (RMSEA) = 0.098. These indices indicated a good fit with the data; χ^2^/df = 4.213 suggested the model had an acceptable validity [44]. Thus, the model was considered appropriate.

An analysis of the TITC-T with CFA resulted in a four-factor model. Standardized factor loadings for all items of the four factors ranged from moderate to strong: 0.78 to 0.91 for the factor of constructive controversy; 0.68 to 0.88 for helping behaviors; 0.64 to 0.90 for the factor of spontaneous communication; and 0.77 to 0.89 for the fourth factor of team creativity. An examination of the factor loadings suggested that the TITC-T was strongly related to the constructs of constructive controversy, helping behaviors, spontaneous communication, and team creativity with item one (λ = 0.87) and item two (λ = 0.85) for constructive controversy, item three (λ = 0.97) and item four (λ = 0.90) for helping behaviors, item five (λ = 0.93) and six (λ = 0.85) for spontaneous communication, and item seven (λ = 0.84) and eight (λ = 0.90) for team creativity (Figure 1).

### 3.3. Internal Consistency Reliability

The reliability of the TITC-T, as determined by Cronbach’s alpha, was 0.98. The Cronbach’s alpha coefficients for the domains of constructive controversy, helping behaviors, spontaneous communication, and team creativity, were 0.90, 0.95, 0.94, and 0.95, respectively. The results indicated good internal consistency reliability for the TITC-T.

## 4. Discussion

Examining the role of interactions and creativity among team members in the context of collaboration in higher education in Taiwan, especially nursing education, can help one understand and broaden the discussion about interdisciplinary learning [1,3,13,45]. However, a reliable and valid instrument that measures interactions and creativity and specifically targets healthcare populations is not available in Taiwan. Therefore, we developed a measurement instrument for team interactions and team creativity, which was designed for use with Taiwanese nursing students. The psychometric properties of the TITC-T were evaluated with CFA. The TITC-T had a strong internal consistency reliability, as demonstrated by the Cronbach’s alpha for all domains and an acceptable construct validity as demonstrated by CFA, which confirmed the one-dimensional structure. The CFA also confirmed our proposed instrument’s construct validity (χ^2^/df = 4.213, RMSEA = 0.098, NFI = 0.985, CFI = 0.995, IFI = 0.995).

CFA is a theory-driven method; thus, CFA was used to determine if the proposed model identified from the TITC-T fit the data for nursing education. Previous research has established that constructive controversy, helping behaviors, and spontaneous communications influence team creativity and innovation. Our results are comparable to the instruments used in other studies [8,21,32] and have the same results. For instance, Bastian et al. [8] and Marlow et al. [17] showed that providing opportunities for communication and sharing among team members can increase supportive interactions and creativity. Xiang et al. [21] and Derdowski et al. [37] revealed that the interactive behavior of constructive controversy can improve individual and team performance and increase creativity. Therefore, there are correlations among constructive controversy, helping behaviors, spontaneous communication, and creativity. Team interaction and creativity require multiple domains to assess learning effectiveness. The TITC-T is an appropriate assessment tool.

Compared with some existing instruments that assess team interactions and team creativity targeting at the Chinese-speaking populations [14,21], the current study employed forwards- and backwards-translation as recommended by Brislin [40] for the translation of several items of the TTCT-T instrument, which can overcome obstacles of cultural or linguistic differences in the original text. Our study reported significant positive associations between scores for team interactions and team creativity. However, it remains unclear as to whether strong team interactions increased team creativity, vice versa, or both. Longitudinal studies are recommended to examine the direction of causality.

### 4.1. Limitations

This study had some limitations. The TITC-T is a self-report instrument, which assessed nursing students’ perceived abilities and not their true abilities. Studies including participants’ subjective assessments as well as measures that are objective, including the grade from the final team presentation and more objective assessments of creativity are recommended. Second, although our findings would be more compelling if we were able to demonstrate the superiority of the TITC-T scale with another similar measure of team interactions and team creativity, our instrument was developed specifically for nursing students in a novel teaching course, and no similar instrument is in use in Taiwan. Third, most of the nursing students in our study were female (83%). Although this is a reflection of the small percentage of male nurses in Taiwan, it may prevent generalization to other Chinese-speaking cultures with a more heterogeneous representation of nurses. Finally, the participants were recruited from one university, which may also prevent generalization of our findings to other areas of Taiwan.

### 4.2. Implications for Nursing Education

Assessing team interactions and team creativity among nursing students can improve IDE in undergraduate healthcare education programs consisting of interdisciplinary teams, which may improve team creativity [1,36]. The validation and reliability of the TITC-T suggest this could be a promising instrument for nurse educators who wish to have insight into measures of team interactions and team creativity for interdisciplinary nursing student teams in Taiwan and possibly other Asian nursing programs. The TITC-T could also help nurse educators gain an understanding of areas in team interactions that need improvement. Strategies that increase interactions could increase collaboration and team creativity during the development of patentable healthcare-related products.

## 5. Conclusions

This is the first study to develop and validate a self-report scale for measuring team interactions and creativity among nursing students in Taiwan. The results from the CFA and cross-validation conducted in this study indicated that the new TITC-T instrument is psychometrically sound, valid, and reliable. The short statements of the 34-item instrument make it easy to complete. The TITC-T can serve as an assessment tool to identify the strengths of nursing students’ interactions and creativity when working and collaborating in interdisciplinary teams whose goal is to develop patentable healthcare products. In the future, we will promote the TITC-T to be tested and compared by other countries.

## Figures and Tables

**Figure 1 ijerph-19-07958-f001:**
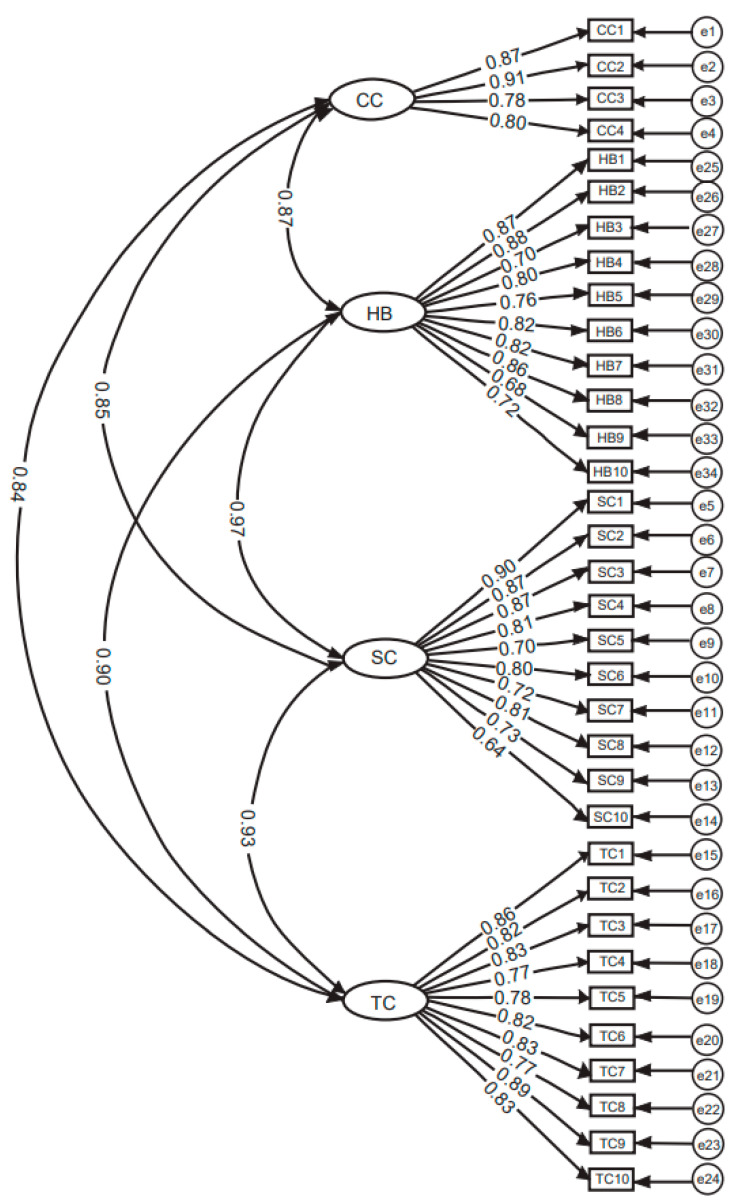
Confirmatory factor analysis of the Team Interactions and Team Creativity scale for Taiwanese nursing students (TITC-T) demonstrated a 4-factor model with standardized factor loadings: constructive controversy (CC), helping behaviors (HB), spontaneous communication (SC), and team creativity (TC). Note: Residual errors (e) indicated as follows: e1 to e4 for CC; e5 to e14 for SC; e15 to e24 for TC; e25 to e34 for HB; CC1 to CC4 denotes the questionnaire items for CC; HB1 to HB10 denotes the questionnaire items for HB; SC1 to SC10 denotes the questionnaire items for SC; and TC1 to TC10 denotes the questionnaire items for TC.

**Table 1 ijerph-19-07958-t001:** Items for the Team Interactions and Team Creativity scale for nursing students in Taiwan (TITC-T).

Construct	Items
**Team interactions**	Members of my team…
Constructive Controversy	1. Encourage me to share thoughts that differ from them.
	2. Listen to diverse perspectives in a supportive manner.
	3. Provide constructive feedback.
	4. Respectful of conflicting thoughts and suggestions.
Helping behaviors	1. Collaborate effectively.
	2. Work efficiently together to complete projects.
	3. Patiently explain new concepts if someone does not understand.
	4. Act as a negotiator if there are conflicting perspectives.
	5. Willingly offer assistance when others need help.
	6. Share the workload equally.
	7. Offer encouragement if a product fails.
	8. Provide constructive suggestions.
	9. Do not hesitate to help others if a problem arises.
	10. Know the team leader will support them.
Spontaneous Communication	1. Are open to exchanging different solutions to a problem.
	2. Are open to discussing new ideas and skills.
	3. Do not hesitate to ask others for information.
	4. Readily share knowledge and skills with other members.
	5. Are available anytime for me to share ideas.
	6. Willing to exchange different methods.
	7. Collaborate with others to stay on schedule.
	8. Offer new methods for old problems.
	9. Encourage discussion to clarify confusing ideas.
	10. Encourage exchanging difficulties as well as successes.
**Team creativity**	1. Have ideas that are new.
	2. Use new techniques to solve problems.
	3. Are able to identify needed healthcare products.
	4. Develop healthcare products that are novel.
	5. Think “outside the box”.
	6. Are good at developing practical solutions.
	7. Can revamp a design to improve it.
	8. Think in ways that are imaginative.
	9. Adds to knowledge and skills to the group.
	10. Can identify a healthcare product that will be useful.

**Table 2 ijerph-19-07958-t002:** Mean scores on the Team Interactions and Team Creativity scale for nursing students in Taiwan (TITC-T) (*n* = 275).

TITC-T Scores	Mean	SD
Team interactions total	4.23	0.48
Subscales		
Constructive controversy	4.00	0.62
Helping behaviors	4.09	0.71
Spontaneous communication	4.05	0.69
Team Creativity total	4.11	0.69

SD = standard deviation.

## Data Availability

The data that support the findings of this study are available on request from the corresponding author, H.-F.C.

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
