# Peer review of "Development and Psychometric Testing of a Taiwanese Team Interactions and Team Creativity Instrument (TITC-T) for Nursing Students"

_ijerph, 2022, doi:10.3390/ijerph19137958_

Round 1

Reviewer 1 Report

Despite some improvements, I'm very bothered by the claim that the proposed method is the first in Taiwan. For an international journal, the first in a particular region is not significant. The authors must have a world-view. If it has been done in another country, the authors must acknowledge and make appropriate comparisons. Otherwise, the manuscript is not publishable. 

Author Response

Thanks for your recommendation. The self-report TITC-T is indeed the first developed assessment tool for nursing education in Taiwan. Since it hasn’t been done in another country, we will promote the TITC-T to be tested and compared by other countries. We have revised the Conclusions (Pages 9, lines 346-347) as follows:

  1. Conclusions

       This is the first study to develop and validate a self-report scale for measuring team interactions and creativity among nursing students in Taiwan. The results from CFA and cross‐validation conducted in this study indicated that the new TITC-T instrument is psychometrically sound, valid and reliable. The short statements of the 34-item instrument make it easy to complete. The TITC-T can serve as an assessment tool to identify the strengths of nursing students’ interactions and creativity when working when collaborating in interdisciplinary teams whose goal is to develop patentable healthcare products. In the future, we’ll promote the TITC-T to be tested and compared by other countries.

Reviewer 2 Report

Thank you for responding to comments and corrections to the article. The comments were comprehensively responded to. Introduced corrections are sufficient corrections. In the current version, the article may constitute material for publication.

Author Response

The authors thank the reviewer for the appreciation and encouragement of our work.

Reviewer 3 Report

The present article could be published in its present form.

Author Response

The authors appreciate the reviewer’s support. 

This manuscript is a resubmission of an earlier submission. The following is a list of the peer review reports and author responses from that submission.

Round 1

Reviewer 1 Report

I'm quite certain there are instruments available already for self reporting on team interaction, and on creativity during team interaction. I'm bothered by the following statement:

"To our knowledge, a measurement instrument which incorporates team interactions and team creativity has not been developed for nursing students in Taiwan."

For an international journal publication, readers are interested in whether or not any instrument is available worldwide. The related work section failed to clarify what existing measurement instruments are available, and if so, what are the issues with these instruments. It appears that the authors side-stepped on this very important issue.

Without comparing with competing instruments, I cannot assess the value of the proposed instrument. Just because the confirmatory factor analysis shows good reliability does not means this is a valuable instrument. 

It would have been much more interesting if the paper could compare the proposed instrument with some competing instruments, and present compelling evidence and analysis on the superiority of the proposed instrument.

Reviewer 2 Report

Thank you for the opportunity to review this article.

The article is interesting, it deals with an issue important for nursing education.

The methodology should be supplemented with a more detailed description of the course of the study, i.e. in which form the questions were asked to the participants? Were they personally asked or was it conducted remotely? In which form did the participants respond?

Figure 1 could have a more detailed description to explain the symbols contained therein e.g. e1, e2 etc.

Reviewer 3 Report

The paper demonstrates development and psychometric testing of a Taiwanese Team Interactions and Team Creativity instrument (TITC-T) for nursing. Regarding to the article my comments are attached. 
